# Dehydration of a crystal hydrate at subglacial temperatures

Alan C. Eaby[1], Dirkie C. Myburgh[1], Akmal Kosimov[2], Marcin Kwit[2], Catharine Esterhuysen[1 ✉], Agnieszka M. Janiak[2 ✉] & Leonard J. Barbour[1 ✉]

Water is one of the most important substances on our planet[1]. It is ubiquitous in its solid, liquid and vaporous states and all known biological systems depend on its unique chemical and physical properties. Moreover, many materials exist as water adducts, chief among which are crystal hydrates (a specific class of inclusion compound), which usually retain water indefinitely at subambient temperatures[2]. We describe a porous organic crystal that readily and reversibly adsorbs water into 1-nm-wide channels at more than 55% relative humidity. The water uptake/release is chromogenic, thus providing a convenient visual indication of the hydration state of the crystal over a wide temperature range. The complementary techniques of X-ray diffraction, optical microscopy, differential scanning calorimetry and molecular simulations were used to establish that the nanoconfined water is in a state of flux above −70 °C, thus allowing low-temperature dehydration to occur. We were able to determine the kinetics of dehydration over a wide temperature range, including well below 0 °C which, owing to the presence of atmospheric moisture, is usually challenging to accomplish. This discovery unlocks opportunities for designing materials that capture/release water over a range of temperatures that extend well below the freezing point of bulk water.

Many crystal hydrates can exchange water with the atmosphere under well-defined conditions of temperature, pressure and relative humidity (RH). They have been classified into three distinct classes[3,4]: ion-associated, isolated and channel hydrates. In channel hydrates the guest water molecules, which can be either stoichiometric or non-stoichiometric with respect to the host, usually form hydrogen-bonded chains and clusters that are loosely associated with the nanoscale pores and thus tend to exchange more readily with their surroundings[5].

Establishing the conditions that govern hydration and dehydration is a critical aspect of materials science. For example, many active pharmaceutical ingredients form hydrates, and spontaneous exchange of water with the surroundings is known to influence their efficacy and long-term stability[6]. Furthermore, the ongoing quest for versatile new materials for desiccation[7] and atmospheric water harvesting[8–10] requires the fine-tuning of several application-specific parameters, of which an important one is a balance between water release kinetics and the energetic cost of thermal regeneration. Hence, the onset temperature $T_{on}$ of water release, that is, the threshold temperature below which the rate of water loss is effectively zero, is a key intensive property of any hydrate[11]. Above $T_{on}$, the rate of dehydration is influenced by environmental factors, sample conditioning and further intensive parameters such as activation energy $E_a$, the frequency factor and reaction order[12]. $T_{on}$ is typically determined by thermogravimetric analysis (TGA) or differential scanning calorimetry (DSC) in an atmosphere consisting of nitrogen purge gas at 0% RH and a pressure of 1 atm[5]. Values of $T_{on}$ for the dehydration of channel hydrates have been reported over the wide temperature range of 20–200 °C (Supplementary Table 1), with $T_{on}$ greater than 60 °C in most cases. Although it is reasonable to suppose that $T_{on}$ is lower than room temperature for materials that undergo humidity-triggered water exchange at room temperature, subambient values of $T_{on}$ are generally not reported. Indeed, their precise values are challenging to determine reliably; $T_{on}$ is typically recorded by heating a sample, and the omnipresence of atmospheric moisture makes it difficult to control the extent of hydration during sample handling at subambient temperatures (see 'Thermal analysis' section in the Methods).

Here we describe a vapochromic channel hydrate that readily and reversibly adsorbs atmospheric water into its 1-nm-wide channels. The stark water-induced colour change allowed us to visually monitor the hydration state of the self-indicating crystals as a function of temperature, and to thus establish unequivocally that the material can release water vapour at temperatures as low as −70 °C. Moreover, on the basis of 122 variable-temperature crystal structure 'snapshots' spanning $T_{on}$, we postulate a mechanism for the sequestration and release of water, which is supported by measurement of the kinetics of dehydration at temperatures between −50 °C and 25 °C. Substantially extending the lower limit of known values of $T_{on}$ has important implications for future tuning of the dehydration kinetics of functional hydrates.

## Host design and crystal packing

Nanoporous molecular solids are difficult to design ab initio because the host building blocks typically pack to optimize intermolecular

[1]Department of Chemistry and Polymer Science, Stellenbosch University, Stellenbosch, South Africa. [2]Faculty of Chemistry, Adam Mickiewicz University, Poznań, Poland. ✉e-mail: ce@sun.ac.za; agnieszk@amu.edu.pl; ljb@sun.ac.za

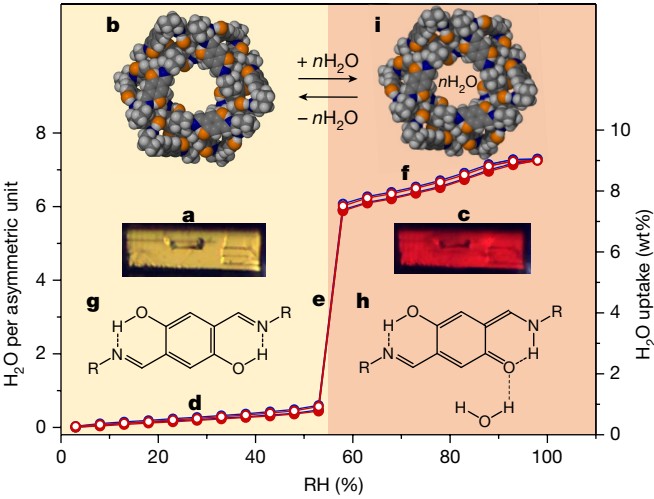

**Fig. 1 | Schematic representation of T1.** 13,27,42,44,45,47-hexahydroxy-3,10, 17,24,31,38-hexaazaheptacyclo[38.2.2.212,15.226,29.04,9.018,23.032,37] octatetraconta-1(42),2,10,12,14,16,24,26,28,30,38,40,43,45,47-pentadecaene.

contacts[13]. Consequently, empty void spaces of greater than 25 Å[3] are rare in molecular crystals[14,15]. Two reliable strategies for creating space for guest diffusion are to use awkwardly shaped host molecules[16,17] or those with intrinsic cavities[18,19]. Rigid macrocycles are good examples of the latter and we have investigated several trianglimines, a class of Schiff-base macrocycles, which pack inefficiently to form discrete cavities or channels[20–22]. Salicylimine moieties were incorporated into T1 (Fig. 1 and Supplementary Text 1) to impose structural rigidity (that is, the intramolecular enolimine hydrogen bonds prevent the aromatic groups from rotating to block the intrinsic cavity). Throughout the text we refer to the trianglimine molecule as **T1**, its yellow, anhydrous form as **T1-Y** (that is, with less than 1 wt% including water) and the red, hydrous form as **T1-R** (greater than 7 wt% water); **T1** refers to either form of crystalline T1. Yellow trigonal prismatic crystals **T1-Y** (Fig. 2a and Supplementary Fig. 1) were grown from an ethanolic solution of T1 (Supplementary Text 2). Our initial single-crystal X-ray diffraction (SCXRD) analysis (Supplementary Text 3) at −173 °C showed that **T1-Y** crystallizes in the trigonal space group *R*3 (Supplementary Table 2). The asymmetric unit comprises two host molecules associated with one another by means of C–H···π contacts. The molecules pack to form extrinsic 10-Å-wide one-dimensional channels that constitute approximately 14% of the volume of the crystal and that propagate along [001] (Fig. 2b and Supplementary Text 3). Only three of the 12 symmetry-independent hydroxyl groups are exposed to the interior of the channel (Supplementary Fig. 4), where they serve as hydrophilic sites for possible host–guest interactions. The crystal structure determined at 25 °C (Supplementary Table 2) is isostructural to that at −173 °C. Low levels of diffuse residual electron density in the channels suggested that the host was almost guest-free, as also confirmed by TGA (Supplementary Fig. 5).

## Vapochromism of T1

Notably, the crystals rapidly and reversibly changed colour from yellow to red in response to increasing RH, with the transition occurring between 53% and 58% RH (Fig. 2a,c, Supplementary Text 4 and Supplementary Video 1). The colour transition occurred as two red fronts emanating from the opposite ends of the acicular crystal and progressing towards its centre along the channel axes [001] and [00Ī]. The slightly slower reverse transition progresses similarly as yellow fronts (Supplementary Video 2). Gravimetric analysis of bulk **T1** confirmed its rapid uptake and release of water, and also showed a dependence of the kinetics on the difference between initial and final RH (Supplementary Text 5.1).

Dynamic vapour sorption (DVS) measurements at 25 °C showed that **T1-Y** gradually adsorbed up to 0.6 wt% of water in the range 3–55% RH (Fig. 2d). In the range 55–58% RH (Fig. 2e), water uptake increased rapidly to 7.3 wt%, after which (Fig. 2f) hydration gradually reached 8.9 wt% at 98% RH. These percentages correspond to non-stoichiometric inclusion

**Fig. 2 | Vapochromic response of T1 to water uptake and release. a**, Micrograph of a single crystal (130 × 100 × 100 μm³) of **T1-Y**. **b**, Corey–Pauling–Koltun (CPK) model of a channel in **T1-Y** viewed along [00Ī]. **c**, Micrograph of the crystal after hydration. **d–h**, Adsorption of water by **T1** at 25 °C: two consecutive DVS isotherms (red and blue) for water uptake below (**d**), at (**e**) and above (**f**) 55% RH. Water-induced tautomerism between the enolimine (**g**) and the ketoenamine (**h**) forms. **i**, CPK model of a channel in **T1-R** viewed along [00Ī].

of 0.5, 5.9 and 7.3 H₂O molecules per host asymmetric unit, respectively. The near absence of hysteresis between the adsorption and desorption profiles implies that the mechanism for water uptake is likely to be the reverse of that for its release. The step at 55% RH corresponding to the colour change thus correlates to the uptake or loss of approximately six water molecules per host asymmetric unit. DVS experiments at 10, 25 °C and 40 °C (Supplementary Text 5.2) yielded almost identical type V[23] isotherms (suggesting weak host–water interactions; $\Delta H = -46 \pm 2$ kJ mol⁻¹—Supplementary Text 5.3). According to Monte Carlo simulations (Supplementary Text 6), a large number of water–water interactions occur at high loadings, which strongly suggests that 'flickering' water clusters[24] and chains connecting the hydroxyl sites may play a role in the rapid sorption of water beyond 55% RH. The colour change from yellow to red is due to a partial shift in the enolimine to ketoenamine tautomeric equilibrium[25–27] as water molecules enter the channel and form hydrogen bonds with the exposed phenolic hydroxyl groups of the host (Fig. 2g,h and Supplementary Texts 7 and 8).

## Structuring of water in the channels

In situ RH-controlled SCXRD analysis was carried out at 25 °C and 80% RH to obtain structural data for the red, hydrous form (**T1-R**) relevant to our observations of RH-driven water uptake at ambient temperature and pressure (Supplementary Text 9 and Fig. 2i). Although it was not possible to model difference electron density peaks unambiguously as ordered chains or clusters of water molecules, we deduced from the diffuse electron density maps that the included water experiences substantial dynamic disorder, but favours interactions with the exposed hydroxyl groups (Fig. 3a). SCXRD analysis of **T1-R** was repeated at −173 °C in the hope of identifying the probable positions of the water molecules in the channel. These data also resisted an unambiguous model for long-range ordered water; the difference electron density maps indicate that water molecules are dispersed throughout the channel, with the highest concentration of electrons still located along the channel walls (Fig. 3b) and within hydrogen bonding distances of the exposed host hydroxyl groups. Although the water loading of **T1-R** at 25 °C was similar to that at −173 °C, the electron density distribution in the guest-occupiable space was far less diffuse at the lower temperature. This suggests that water exists in a state closer to that of a quasi-liquid at higher temperatures

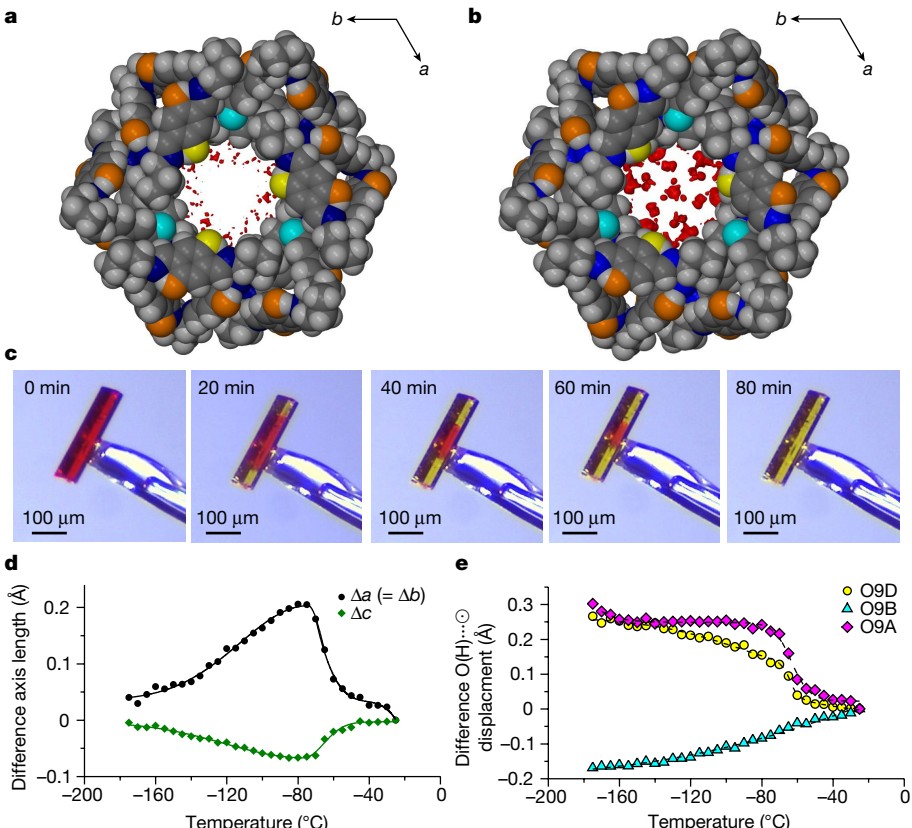

**Fig. 3 | Low-temperature water structuring and release. a,b**, Projections along the channel axis [001] of **T1-R** at 25 °C (**a**) and −173 °C (**b**), indicating that no thermally induced phase changes occur in this temperature range. The concentration and localization of diffuse difference electron density (0.5 e⁻ Å⁻³) on cooling is shown. Colours: red, difference electron density contours; yellow, hydroxyl oxygen O9D; cyan, hydroxyl oxygen O9B; orange, remaining oxygen atoms; grey, carbon; blue, nitrogen; white, hydrogen. **c**, Photomicrographs of an initially hydrous red single crystal of **T1** at −20 °C during chromogenic transition to yellow during dehydration. **d,e**, Difference axis lengths (**d**) and difference O(H)···☉; displacements (**e**) track differences between the hydrous and anhydrous crystals on cooling, referenced with respect to the latter. ☉ represents the centroid of symmetry-equivalent oxygen atoms.

and then accumulates at more well-defined regions on cooling. If water forms long-lived clusters at −173 °C, then its apparent lack of long-range order implies that these quasi-solid aggregates are not commensurate with the periodicity of the host molecules. Note that in addition to the host frameworks of **T1-R** at 25 °C and −173 °C being isostructural, **T1-Y** is the isomorphic dehydrate of **T1-R**[28].

## Dehydration at subglacial temperatures

It is not possible to infer from only the two hydrous crystal structures determined at vastly different temperatures described above whether the included water molecules experience a reversible disorder to short-range-order transition (analogous to freezing and melting) over a narrow temperature range, or if the process occurs gradually on thermal cycling. Therefore, subambient variable-temperature SCXRD (VT-SCXRD) studies were carried out, initially using a cryostat that used dry N₂ gas at 1 bar. Under these conditions, the hydrous crystals still transformed from red to yellow (indicating dehydration), remarkably even at temperatures below 0 °C. Indeed, the colour of a **T1** crystal provides a convenient visual indication of its hydration state, which would otherwise be difficult to ascertain at low temperatures. Variable-temperature optical microscopy (Fig. 3c and Supplementary Videos 4 and 5) showed that the crystals remained red for longer than 5 days at temperatures below −70 °C, whereas a distinct change to yellow indicated water loss at temperatures above this (Supplementary Text 10).

VT-SCXRD data (Supplementary Text 11) were recorded at intervals of 5 °C for **T1-R** from −25 °C to −175 °C, and then back to −25 °C. A similar series of diffraction experiments were carried out for **T1-Y**, and we

thus obtained 122 structural 'snapshots' comprising 61 isothermal pairs of structures of both the hydrous and anhydrous forms. As it is an isomorphic dehydrate of **T1-R**, **T1-Y** was used as a control to discern subtle temperature-dependent structural changes associated with the immobilization of included water on cooling. We monitored changes in the difference electron density maps due to water in the channels (Supplementary Videos 8–11). In the range from −25 °C to −50 °C, the densest electron clouds of the hydrous crystal congregate near the host hydroxyl groups, but from their diffuse distribution (Fig. 3a) we infer that the water molecules are still highly mobile. Further cooling locks the electron clouds into positions close to the hydroxyl groups and also results in a gradual localization of electron density throughout the channel (Fig. 3b). Contrasting the temperature-dependent change in the crystallographic axes (Fig. 3d) and displacement of the three unique exposed hydroxyl groups in the hydrous and anhydrous crystal structures (Fig. 3e) provides molecular-level insight into the behaviour of water in the channels. In the presence of water, the host molecules pivot such that two of the hydroxyl groups (O9A and O9D) gradually migrate outward on cooling, with the onset of an abrupt step occurring at −60 °C. Below −70 °C the gradual outward drift resumes. The remaining hydroxyl group (O9B) gradually migrates inwards over the entire temperature range of −25 °C to −175 °C.

The structural bracing effect (reminiscent of the behaviour of bulk water on freezing) that occurs between −60 °C and −70 °C is probably brought about by non-lattice ordering of the water in **T1-R** on cooling, which also inhibits the transfer of water to the surroundings. Indeed, it is known that the dehydration temperature of a channel hydrate depends on the level of geometric frustration imposed on the included water by the

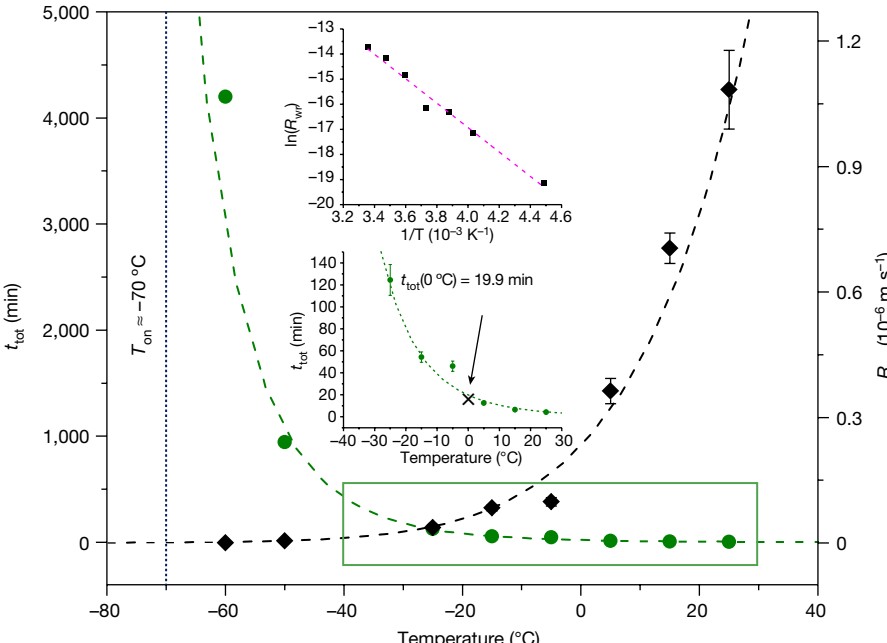

**Fig. 4 | Low-temperature dehydration kinetics.** Plot of the total dehydration times $t_{tot}$ (green, left ordinate) and $R_{wr}$ (black, right ordinate) as a function of temperature for a single crystal of length 272 μm along [001]. The dashed lines correspond to the Arrhenius fit of the data points. The dotted vertical line indicates the estimated value of $T_{on}$. Insets: Top, Arrhenius fit of the zero-order reaction model of channel emptying in **T1**. The least squares regression line of best fit is shown in red. Bottom, magnified temperature range, as indicated by the green rectangle, for $t_{tot}$ as a function of temperature. The cross indicates $t_{tot}$ at 0 °C. Notably, the observed $t_{tot}$ at −60 °C is larger than that predicted by the Arrhenius parameters. However, this is not surprising considering that the onset of geometric frustration occurs at this temperature. Note that the kinetic data presented here are relevant to the crystal used, and would vary depending on the size of the crystal[34]. Error bars represent the standard error of the mean of three measurements.

host framework[29]. That water can flow within the crystal at temperatures above −70 °C is probably due to a mismatch between the geometry of the channel (that is, its diameter and the locations of the exposed hydroxyl groups) and a hypothetical long-range ordered ice-like structure[30]. Our structural study thus suggests that cessation of dehydration at temperatures lower than −70 °C may be related to short-range ordering of the included water, akin to freezing or a glass transition. Indeed, water confined to narrow channels (less than 2 nm) is known to assume an amorphous glass-like state when geometric constraints prevent it from forming distinct networks with long-range order[31]. These structural results are supported by DSC measurements (Supplementary Text 12).

## Low-temperature dehydration kinetics

Because the included water is mobile above −70 °C and the colour change associated with dehydration is uniform along the channel axis, it was possible to investigate the dynamics of pore emptying in **T1** across a wide temperature range using optical microscopy. The rate of channel emptying was determined by measuring the evolution of the yellow boundaries during dehydration at selected temperatures ranging from 25 °C to −50 °C (Supplementary Fig. 59). At each temperature, water release (wr) occurs at an approximately constant rate $R_{wr}$, implying a zero-order reaction mechanism (Supplementary Text 13.1).

The rate of channel emptying follows an Arrhenius temperature dependence (Fig. 4) with an activation energy of $E_{a,wr} = 41 \pm 2$ kJ mol⁻¹. It is important to note that activation energies for dehydration are known to decrease with decreasing particle size distribution; since our study necessarily involved a relatively large single crystal, the activation energy for dehydration of **T1** is likely to be slightly lower than the measured value of 41 kJ mol⁻¹. This value is significantly lower than that for the sublimation of ice (53–58 kJ mol⁻¹)[32,33], which is also a zero-order process. It is also much lower than those reported for thermal dehydration (that is, heating above room temperature) of crystal hydrates at

1 atm (Supplementary Table 16), but comparable to those for thermal dehydration of silica gels under vacuum. The relatively low activation energy for water release by **T1** in the range 25 °C to −50 °C is thus consistent with the ability of **T1** to dehydrate even at subglacial temperatures. The rate of water release below −70 °C is difficult to measure; extrapolation of the data in Fig. 4 suggests that $R_{wr} = 6.3 \pm 0.4 \times 10^{-10}$ m s⁻¹ at −70 °C, which would require approximately five days for the selected crystal to dehydrate completely. However, even after five days at −70 °C, some regions of the crystal were still red, implying changes in the flow dynamics due to vitrification of the water.

The ease of water transport through **T1** was assessed using mean square displacement (MSD) analysis (Supplementary Text 13.2). A diffusion coefficient $D$ was determined in silico by monitoring the displacement of water molecules in a hydrous crystal at different temperatures $T$. These calculations show that the nanoconfined water in **T1** experiences higher rates of diffusion than in other materials, both above and below the normal freezing point of bulk water (Supplementary Table 17). The diffusion rates of water through **T1** at both 25 °C ($D_{T1} = 10.6 \times 10^{-9}$ m² s⁻¹) and −53 °C ($D_{T1} = 1.7 \times 10^{-9}$ m² s⁻¹) are approximately four times those for its diffusion through bulk water at 27 °C ($D_{Bulk} = 2.6 \times 10^{-9}$ m² s⁻¹)[34] and ice at −53 °C ($D_{Ice} = 0.35 \times 10^{-9}$ m² s⁻¹)[35], respectively. Moreover, the diffusion of water through **T1** also follows an Arrhenius-type temperature dependence, with an activation energy of $E_a = 15 \pm 1$ kJ mol⁻¹ (Supplementary Text 13.2 and Supplementary Fig. 71), which is lower than that for water release, implying that channel emptying is the rate determining step for dehydration.

## Discussion

Crystal hydrates must often meet strict, application-specific criteria regarding their chemical composition and micromeritic properties. Because these factors influence the dehydration kinetics, it is important to determine the total dehydration time $t_{tot}$ (hydrous to anhydrous)

of a hydrate. For instance, practical $t_{tot}$ regeneration times are usually obtained by heating desiccants to well above $T_{on}$, albeit at a potentially high energy cost. Striking the optimal balance between dehydration times and associated energy costs requires detailed knowledge of the kinetics of dehydration, in particular $T_{on}$ and the Arrhenius parameters (as determined for the material in the form relevant to its intended purpose). By definition, $t_{tot}$ tends to infinity as the dehydration temperature approaches $T_{on}$ during cooling. Despite the general difficulty of studying dehydration at subglacial temperatures, we have demonstrated that $T_{on}$ can be unexpectedly low and that $t_{tot}$, even for the relatively large particles studied here, can be in the order of minutes at temperatures well below the normal freezing point of water; by interpolation of the plot of $t_{tot}$ versus temperature for **T1-R** (Fig. 4, inset), using the Arrhenius parameters, $t_{tot}$ at 0 °C is 19.9 min for the crystal studied. Under the same set of conditions of temperature, pressure and percentage of RH, reducing the size of the particles would reduce $t_{tot}$[36], but not $R_{wr}$. We note that kinetic parameters are typically not reported for hydrates, particularly those exhibiting subambient water desorption[2,27,37] (Supplementary Text 14).

## Conclusion

We have established visually that $T_{on}$ for dehydration of **T1** is close to −70 °C. Variable-temperature X-ray diffraction studies suggest that the included water undergoes a reversible structuring event at this temperature. Above $T_{on}$, water appears to experience liquid-like mobility in the channels, but cooling below $T_{on}$ initiates a sudden concentration of the water molecules at the hydroxyl binding sites that line the channels. This probably involves local ordering, accompanied by a pronounced bracing effect (due to either the formation of aperiodic water clusters or vitrification of water), as evidenced by a net outward force being exerted on the channel by the included water. The lack of long-range order of the water molecules below $T_{on}$ indicates that the hydroxyl binding sites of the host channel are not optimally positioned to support a well-defined arrangement of water clusters and/or chains, and this structural mismatch between host and guest probably points to an essential design requirement for tuning low values of $T_{on}$. The implications of this finding are relevant to studies of other classes of channel hydrate for applications in which tuning the onset temperature of water release is critical. Channel hydrates that form isomorphic dehydrates include active pharmaceutical ingredients and other molecular solids, as well as the rapidly growing assortment of rigid hygroscopic metal-organic frameworks and covalent organic frameworks. A cursory search of the literature reveals some examples of other materials that may also exhibit $T_{on} < 0$ °C (Supplementary Text 14). Of these, metal-organic frameworks and covalent organic frameworks are particularly amenable to synthetic and structural fine-tuning. To our knowledge, this work represents one of the first systematic studies of the kinetics of subambient dehydration at 1 atm, and establishes a benchmark value of $T_{on} \approx -70$ °C for low-temperature water release from a crystalline hydrate, thus extending the lower range of the continuum of temperatures at which dehydration can be achieved.

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

## Methods

### Materials

All commercially available reagents were obtained from commercial suppliers and used in reactions without further purification unless otherwise specified. The anhydrous dichloromethane and chloroform were distilled over calcium hydride under inert atmosphere. The anhydrous teterahydrofuran was distilled over potassium metal with benzophenone as indicator. Flash column chromatography was performed using Merck Kieselgel type 60 (250–400 mesh). Merck Kieselgel type 60 F$_{254}$ analytical plates were used for thin-layer chromatography. $^1$H and $^{13}$C nuclear magnetic resonance (NMR) spectra were recorded on a Bruker 300-MHz or Bruker 400-MHz spectrometer at ambient temperature. The $^1$NMR spectra are reported in parts per million downfield of tetramethylsilane and were measured relative to the residual signals for CDCl$_3$ (7.27 ppm and 77.0 ppm, respectively, for $^1$H and $^{13}$C NMR). The $^{13}$C NMR spectra were obtained with $^1$H decoupling. Mass spectra were recorded on AB Sciex TripleTOF 5600+ System. Melting points were measured using open glass capillaries in a Büchi Melting Point B-545 apparatus. A Jasco P-2000 polarimeter was used for optical rotation measurements (at 20 °C). Infrared spectra were measured using a Thermo Scientific Nicolet iS50 FTIR spectrometer.

### Scanning electron microscopy

High-resolution images were recorded on the Quanta FEG-250 Scanning Electron Microscope using an accelerating voltage of 2 kV and a working distance of 7.8 mm. Images of an anhydrous crystal fixed to a holder with carbon tape were recorded under high vacuum ($3.25 \times 10^{-3}$ Pa).

### Single-crystal X-ray diffraction

Data were recorded using a Bruker D8 Venture equipped with a PHOTON II CPAD detector and an Oxford Cryosystems Cryostream 800 Plus cryostat. MoKα X-rays (λ = 0.71073 Å) were generated using a multilayer Incoatec Microfocus Source (IμS) . A crystal-to-detector distance of 37 mm was used for all experiments. Data reduction was carried out using the Bruker SAINT[38] software. Absorption and other corrections were made using SADABS[39] as implemented in the Bruker APEX 3 software package. Crystal structures were solved either by means of direct methods using SHELXS[40] or by intrinsic phasing with the SHELXT[41] through the X-Seed[42,43] graphical user interface. Non-hydrogen atoms of the host were refined anisotropically using SHELXL[44], using full-matrix least squares minimization. Host hydrogen atomic positions were calculated using riding models. The absolute structure of the investigated crystals was assumed from the known absolute configuration of the (R,R)-1,2-diaminocyclohexane, which was used as a starting material in the syntheses. The strategy used to prevent water loss during VT-SCXRD experiments is described in Supplementary Text 11.

### Crystallographic software

Probe-accessible extrinsic channel and intrinsic cavity volumes were calculated using MSRoll[45], using a probe radius of 1.5 Å, and visualized using Mercury[46]. Difference electron density maps were calculated using Marching Cubes 2005[47] as implemented in CRYSTALS 15.0.1 (ref. [48]), and rendered using Persistence of Vision Raytracer[49]. The Cambridge Structural Database[50] (v.5.42, database: May 2021) was accessed using ConQuest[51] and geometrical data characteristic of enolimine and ketoenamine tautomers were assessed using Mercury.

### Ultraviolet–visible spectroscopy

Solid-state ultraviolet–visible spectra were recorded using an Analytik Jena Specord 210 Plus (Edition 2010) spectrophotometer equipped with an integrating sphere for diffuse reflectance measurements. The instrument uses a combination of halogen and deuterium light sources, with the lamp switchover set to 400 nm. Samples were placed onto the sample stage and the integrating sphere was flushed with dry N$_2$ gas to dehydrate the sample. Spectra were measured in absorption mode over a wavelength range of 400–900 nm. High humidity conditions were simulated by bubbling the N$_2$ gas through water for 30 minutes before data acquisition. Spectra were analysed using the WinASPECT PLUS software package.

### Attenuated total reflectance Fourier transform infrared spectroscopy

Attenuated total reflectance Fourier transform infrared spectroscopic (ATR-FTIR) analysis was carried out using a Bruker Alpha P FTIR infrared spectrometer equipped with a QuickSnap Platinum ATR module. Data were acquired using the OPUS 15 (Bruker, v.7.5) program and processed in Microsoft Excel.

### Simulating ultraviolet–visible spectra

Time-dependent density-functional theory calculations were performed with the Gaussian 09 (ref. [52]) software suite using functionals from a range of classes, namely local (B97D (refs. [53,54]), B98 (ref. [55]) and PBEPBE-D3 (ref. [56])), hybrid (B3LYP-D3 (ref. [57])), meta (M06-D3 (ref. [58])) and long-range corrected (CAM-B3LYP-D3 (ref. [59]) and ωB97XD (ref. [60])) functionals. Grimme's D3 dispersion corrections[54,61,62] were added to all functionals except for B98, for which GD3 has not been parameterized. B3LYP-D3 was used for all other calculations in this study. Geometry optimizations were performed with the Berny[63] optimization algorithm.

Only Gaussian-type[64] orbital basis sets were used: the 6-311++G(d,p) basis set[65–67] was used for calculations done on the model chromophore, whereas the 6-311G basis set was used for the full trianglimine molecule. Frequencies were calculated at the same levels of theory to confirm that minimum-energy structures had been obtained from the optimizations. An implicit solvent model, the self-consistent reaction field[68] approach with the polarizable continuum model, was used to model a water ($\varepsilon = 78$) environment when studying the interactions between the chromophore and explicit H$_2$O molecules.

### Preparation of structures of T1 for molecular mechanics calculations

All molecular mechanics calculations were carried out using the BIOVIA Materials Studio (MS) 2018 software suite[69]. Each structure was prepared for further computational analysis using the following optimization sequence: non-hydrogen atomic positions were taken from the low-humidity experimental crystal structure determined from high-quality SCXRD data at −173 °C, unless otherwise specified. The positions of the enolimine hydrogens were fixed to an idealized intramolecular hydrogen bond. The remaining hydrogen atoms of the framework were optimized as part of a periodic system using the Forcite module of the MS software suite. The Smart algorithm[70,71] was used with a convergence tolerance of $2 \times 10^{-5}$ kcal mol$^{-1}$, a maximum force of $1 \times 10^{-3}$ kcal mol$^{-1}$ Å$^{-1}$ and displacement of $1 \times 10^{-5}$ Å$^{-1}$. The Condensed-phase Optimized Molecular Potentials for Atomistic Simulation Studies (COMPASS II)[72] force field was used with the charges automatically assigned. COMPASS II partial charge assignments and valence parameters are derived from ab initio calculations and optimized to be consistent with experimental data[73]. The electrostatic and van der Waals interactions were summed using the Ewald and atom-based methods, respectively.

### Simulating sorption of water

Sorption in **T1** was simulated using the Sorption module with the COMPASS II force field and the charges automatically assigned. The solvent-accessible regions were calculated using a Connolly radius of 1.5 Å and a grid interval of 0.15 Å. The simulations ran for $1 \times 10^6$ equilibration steps and $1 \times 10^7$ production steps at 25 °C. The Adsorption Isotherm task, which makes use of the grand canonical thermodynamic ensemble, was used to simulate the adsorption of H$_2$O molecules at a fixed fugacity; the number of sorbate molecules

was varied until equilibrium was reached in a series of fixed pressure runs. The Fixed Loading task, which makes use of the canonical thermodynamic ensemble, was used to identify the probable positions of water by fixing the number of $H_2O$ molecules, the unit cell volume and simulation temperature. $H_2O$ molecule adsorption configurations were sampled by the Metropolis Monte Carlo method[74], which filters allowable transformations. Trial configurations were generated without bias and transformations that resulted in a state with a higher probability were accepted, whereas others were rejected. Trial states were governed by the force-field-derived potential energy. The minimum energy sorbate positions were visualised in the Visualizer module of MS and the superposition of degenerate states compared as probability density maps.

## Molecular dynamics

Molecular dynamics (MD) simulations were carried out using the Forcite module with the COMPASS II force field and automatically assigned charges. The NVT (a fixed number of atoms, N; a fixed volume, V; and a fixed temperature, T) thermodynamic ensemble was used on periodic structures that had been loaded with $H_2O$ molecules using the Sorption module. The initial velocities were randomized over 10 ns of simulation time at a 1.0-fs time step and the simulation temperatures were controlled using the Nosé–Hoover–Langevin thermostat[75]. The results obtained from MD calculations were analysed using the Forcite Analysis dialog. Specifically, the radial distribution function and MSD analyses were used in this work.

Radial distribution function analysis: the number of $H_2O$ molecules was systematically increased from one to six water molecules per channel in the unit cell. Each simulation provided 10,000 frames in which the lengths of the hydrogen-bond interactions were identified by monitoring the water–water interactions and water–OH group interactions up to a maximum pair distance of 0.6 nm.

MSD analysis: the non-hydrogen atoms were obtained from a high-quality SCXRD structure measured at −50 °C. A total of 20 water molecules per channel were loaded using the Fixed Loading protocol in the Sorption module and MD was performed at temperatures between −173 °C and 25 °C, as previously described. The MSD analysis was carried out on the first 1.5 ns of the MD calculations.

## Thermal analysis

TGA was carried out using a TA Instruments Q500 thermogravimetric analyser. The instrument measures the change in sample mass as a function of temperature. The sample was heated at a rate of 10 °C min⁻¹ under a dry $N_2$ gas purge of 60 cm³ min⁻¹ from room temperature to 80 °C. Thermograms were analysed using the TA Instruments Universal Analysis program.

It is technically challenging to determine a subambient $T_{on}$ using TGA, which is usually carried out by heating from room temperature; standard TGA instruments are not configured for low-temperature analysis. If available, a cooling unit would cause any atmospheric water present to condense on the sample and balance components during sample loading. Although condensation can be avoided by carrying out experiments under dry conditions, this solution has its own limitations: materials with a low $T_{on}$ will dehydrate rapidly in a 0% RH environment.

DSC thermograms were recorded using a TA instruments Q100 analyser equipped with a liquid nitrogen cooling system. During experiments, the sample compartment was purged using either nitrogen (50 cm³ min⁻¹) or helium (25 cm³ min⁻¹) gas. Further details of the experiments are given in Supplementary Text 10.

## Dynamic vapour sorption

DVS analysis was used to quantify the water uptake by porous **T1** at 10 °C, 25 °C and 40 °C. These measurements were carried out using a DVS Advantage analyser (Surface Measurement Systems Ltd) with nitrogen as the carrier gas. A powdered sample (c. 6 mg) was loaded onto the balance pan of an analyser cell and activated at 0% RH and 40 °C for 24 h. On completion of activation, the sample compartment was equilibrated to the specified temperature of measurement (10 °C, 25 °C or 40 °C) and the RH was increased from 3% to 98% in 5% steps, and then decreased to 3% in 5% steps. During each step, the RH was maintained until the mass change was less than 0.001 wt% min⁻¹. The full sorption–desorption cycle was carried out twice at each temperature. To show that the sudden increase in water uptake at 55% is not due to large humidity increments, an experiment with a narrow humidity range was performed at 25 °C for a fresh sample of 4.6 mg. The sample activation parameters were maintained as for the full-range experiment, whereas the measurement of water uptake was started in the humidity cycle range of 43–58% in 1% increments.

## Low-temperature differential scanning calorimetry

DSC thermograms were recorded for hydrous crystals during cooling and heating between −2 °C and −160 °C (Supplementary Text 12). To increase the sensitivity of the measurements, a sample consisting of 4.05 mg of single crystals was placed directly onto the DSC sensor stage (Supplementary Fig. 54).

## Gravimetric water sorption kinetics

Water vapour sorption experiments were carried out using a sorption balance system developed in-house[76]. Saturated salt solutions were used to maintain the desired RH conditions. Adsorption data were recorded from 11% RH (LiCl solution) to the target value, followed by desorption from the target value back to 11%. For a typical experiment, the balance was tared with an empty pan at 11% RH. The sample was then loaded onto the pan and its weight recorded at 11% RH, after which the balance was tared again. A sorption experiment was initiated by replacing the LiCl solution with a solution that delivers a higher RH. The weight was recorded as a function of time and data were scaled to yield $\alpha$–time plots in the range $\alpha = 0$ to 1, where $\alpha = 0$ represents the weight at $t = 0$ and $\alpha = 1$ represents the weight at equilibrium. Rate constants were determined using deceleratory kinetic models[77].

## Kinetics measurements using optical microscopy

A similar method to that of Takamizawa[78], who studied the dynamic flow of included molecules in a single crystal using optical microscopy, was used to determine the rate of channel emptying. In a typical experiment, a crystal of **T1-Y** was glued to a glass fibre, converted to **T1-R** by exposure to high humidity and then placed in the dry nitrogen stream at the experimental temperature. Images of the crystal were recorded at regular intervals and the lengths of the advancing yellow regions $d$ were plotted as a function of time $t$ (Supplementary Figs. 60–66).

## Data availability

The crystallographic data are archived at the Cambridge Crystallographic Data Centre under reference numbers CCDC 2126755–2126880. All other data used in this study are available from the corresponding authors on reasonable request. Source data are provided with this paper.

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

**Acknowledgements** C.E., A.C.E. and D.C.M. thank the Centre for High-Performance Computing for access to computation resources. The authors also acknowledge J. Steed, E. Strauss and T. Friščić for insightful comments. This work was supported by the National Research Foundation and Stellenbosch University.

**Author contributions** M.K. and A.K. synthesized the trianglimine. A.M.J. crystallized the material, carried out the initial crystal structure determinations, and performed the DVS and SEM analyses. C.E. led the computational analyses, which were carried out by D.C.M. and A.C.E. A.C.E. carried out optical microscopy, TGA, DSC, gravimetric kinetic measurements and VT-SCXRD. L.J.B. led the experimental efforts and co-wrote the manuscript with A.C.E.

**Competing interests** The authors declare no competing interests.

**Additional information**
**Correspondence and requests for materials** should be addressed to Catharine Esterhuysen, Agnieszka M. Janiak or Leonard J. Barbour.
