## [Peer Review File · Nature]

Manuscript Title: Dehydration of a crystal hydrate at sub-glacial temperatures

Reviewer Comments & Author Rebuttals

Reviewer Reports on the Initial Version:

Referees' comments:

Referee #1 (Remarks to the Author):

The authors report the synthesis of a porous molecular organic crystal "T1" that can reversibly adsorb water into 1 nm channels (up to about 9 wt%), lined with some hydroxyl groups, at moderate relative humidity. As in many solvatochromic porous materials, the water sorption is accompanied by a color change (transition between 53 and 58% RH), which can be used to monitor uptake over a wide temperature range. The authors used a combination of different techniques (XRD, microscopy, DSC and molecular simulations) to ascertain that the confined water above -70 °C exists in a state of flux/disorder, which helps to allow dehydration of the hydrate to be observed due to weaker interactions than in bulk water.

Type V isotherms were observed, also pointing to weak host-water interactions. SCXRD data did not yield evidence for an ordered included water phase, but not surprisingly, enhanced electron density was found near the hydroxyl groups in the channels. Moreover, the kinetics of dehydration were determined down to temperatures below 0 °C – but does this not depend strongly on diffusion and thus on the size of the crystals? The Arrhenius-type activation energy for dehydration was found to be at the order of 40 kJ/mol, similar to that for the dehydration of silica gels.

Comments:

It is not clear why the authors find it particularly difficult to monitor desorption at lower temperatures – all that is needed to exclude atmospheric humidity is to perform the experiments under controlled gas atmosphere.

T(on) itself is not well defined – how long will you wait to see first traces of water emerge from the material? And which crystal size will you take as the reference size? For example, it has been shown that even hydrophilic zeolites can be dried at room temperature (instead of several hundred degrees) within short times when the crystals are present at the nanoscale. I would therefore argue that T(on) is by no means an intrinsic property of the materials – it strongly depends on size!

Was the enthalpy of adsorption in the host determined?

It is not clear why the authors did not perform their VT-SCXRD studies under equilibrium conditions at low partial pressures, to prevent water loss during experiments.

The authors completely miss a discussion of the vast literature on water sorption and desorption in zeolites, mesoporous solids, MOFs and COFs.

Summarizing, the authors have shown that a channel host can absorb and desorb water, down to low temperatures. Such processes critically depend on the enthalpy of water ad/desorption, which can be controlled within a vast range via structure and functionalization of the host, and kinetically based on the physical size of the host material. There is really nothing surprising or original here. Vast numbers of other materials exist whose interactions with water can be precisely tuned to reach similar behavior (the authors cite silica gel), including MOFs and COFs. It is surprising that the authors do not discuss in detail such other materials in their paper. As a result, the importance and potential impact of this work are far below the level expected for a paper in Nature. Publication is not recommended.

Referee #2 (Remarks to the Author):

This is a very interesting example of water uptake and loss from a smartly designed nanoporous material. The experimental work is meticulously done. The text is readable and figures are gorgeous. It is really very challenging to find anything to criticize.

So that you know I read it, on page 7 line 9, there should be an "is" before "also"

Though not a requirement for publication, it could be interesting to look at D₂O / H₂O exchange in these materials perhaps under Raman microscopy conditions. That would provide an alternative way to estimate water diffusion rates and mobilities in/out of the pores.

This should be published.

Referee #3 (Remarks to the Author):

Manuscript# 2022-04-05193A-Z describes the synthesis and characterization of a molecular solid that crystallizes to form extrinsic channels, as well as intrinsic cavities. The molecular design includes pendant hydroxyl groups, such that some (3 of the 12 symmetry-independent OH groups) are exposed in the channels and thus available for hydrophilic interactions with guest molecules, presumably water and presumably via H-bonding. The molecular design also includes enolimine moieties that are in equilibrium with their tautomeric ketoenamine form, and this equilibrium can be shifted by the introduction/removal of guest water molecules. The authors have carefully characterized the water uptake and release from a representative range of their crystals. They have employed variable-temperature structural analysis via X-ray diffraction, TGA measurements, dynamic vapour sorption analysis – including kinetic studies, appropriate computational analysis, and visual tracking courtesy of a red/yellow colour change associated with the shift in keto/enol equilibrium as a function of water inclusion.

This work is presented clearly. Discussion is clear and lucid. Statements and conclusions are supported by the results. The data quality appears to be good. The methodology appears to be appropriate, although perhaps incomplete (see questions/suggestions below).

The extraordinary aspect of the work presented in this manuscript is the observation and characterization of sub-ambient T(on) values, indeed well-below the freezing point of bulk water. The implication is that this specific material, and the specific characterization techniques employed herein, significantly push the boundaries of our water absorption/release technology.

In general, I believe this work to be worthy of publication in Nature. I do have some suggestions, however:

- P4, line 20: The fact that the crystal structure determined at 25 degC is isostructural to that at -173 degC does not rule out the possibility of a re-entrant phase transition.
- I am surprised that solid state ¹H NMR was not used to investigate this system, either for the enolimine to ketoenamine tautomeric equilibrium, or for the uptake/release of water, or for other characterization of the hydroxyl groups. Was this methodology considered and dismissed for some reason or was it overlooked?
- I am less surprised that spectroscopy in the IR spectral range was not employed (e.g., absorption or reflectance FT-IR; Raman), since interpretation might be more difficult. However I am wondering whether this was an omission by chance or whether the possibility of tracking O-H stretching frequencies and H-O-H bending/distortion modes was not considered valuable enough to attempt to collect and interpret data. Had the authors considered and dismissed the idea?

Author Rebuttals to Initial Comments:

RESPONSE TO REFEREES

We thank all three referees for their appreciation of our work, as well as their insightful comments/suggestions. Below we provide detailed responses to each of the points that they have raised (*their text in blue*), and that should be addressed.

Referee #1

(summary of the manuscript omitted)

Moreover, the kinetics of dehydration were determined down to temperatures below 0 °C – but does this not depend strongly on diffusion and thus on the size of the crystals?

Response: our detailed response is given further below, where the referee expands on this point.

It is not clear why the authors find it particularly difficult to monitor desorption at lower temperatures – all that is needed to exclude atmospheric humidity is to perform the experiments under controlled gas atmosphere.

Response: We presume that the referee is referring to our statement in the abstract “We were able to determine the kinetics of dehydration over a wide temperature range, including well below 0 °C which, owing to the presence of atmospheric moisture, is usually challenging to accomplish”. Water sorption/desorption isotherms are usually recorded using dynamic vapour sorption instruments. The company Surface Measurement Systems is one of the leading suppliers of such equipment to the community of researchers concerned with studies of hydrates. Of the six different gravimetric instruments that they produce for this purpose, none are capable of measurements below 5 °C. Owing to the low vapour pressure of water below its normal freezing point, volumetric sorption systems are also not suitable for subglacial water sorption/desorption measurements. We agree that one can exclude atmospheric water by carrying experiments out under controlled gas atmospheres, which is why we employed temperature-controlled dry nitrogen gas to facilitate our desorption kinetic measurements. However, this approach requires a discernible indication of the hydration state of the material (such as a colour change). Moreover, using this approach a fully hydrated crystal can be exposed to the dry nitrogen stream in a matter of seconds, thus precluding a change in its hydration state during sample handling. This is particularly important when dealing with a sample such as **T1**, which dehydrates rapidly.

T(on) itself is not well defined – how long will you wait to see first traces of water emerge from the material? And which crystal size will you take as the reference size? For example, it has been shown that even hydrophilic zeolites can be dried at room temperature (instead of several hundred degrees) within short times when the crystals are present at the nanoscale. I would therefore argue that T(on) is by no means an intrinsic property of the materials – it strongly depends on size!

Response: It is not clear whether the referee means that (i) the concept of T(on) is not well defined, or (ii) that the determination of T(on) in our specific case is not well defined. We will address both interpretations.

(i) $T(\text{on})$ is a commonly used term in thermal analysis; it is the temperature at which a thermally-induced process (e.g., a phase change and/or release of volatile content) is detectable and its value is usually determined by extrapolation. In the context of host-guest chemistry of solvates, $T(\text{on})$ is considered "a reliable measure of thermal stability" (L. R. Nassimbeni, *Acc. Chem. Res.* **2003**, *36*, 631–637; we have added this reference on p. 2 line 24).

(ii) As the referee suggests, it is almost impossible to determine by direct observation how long it takes for the first water molecule to leave the material. Similar difficulties apply to the direct measurement of many other intensive properties (e.g., melting point, molar enthalpy, etc.) because experimental measurements are influenced by parameters such as particle size distribution, heating/cooling rates, external pressure, guest loading, etc. It is therefore common practice to specify these parameters when reporting such values; we are clear about reporting dehydration of a crystal of dimensions $272 \times 63 \times 83 \mu\text{m}^3$ under conditions that are easily accessible to other researchers (1 atm pressure, 0% RH temperature-controlled nitrogen gas). Kinetic processes that follow Arrhenius behaviour slow down with decreasing temperature. Once the Arrhenius parameters have been determined, the temperature at which the process *effectively* stops occurring (i.e. $T(\text{on})$) can be estimated by extrapolation – as is the case in our report.

As the referee correctly points out, smaller particles are expected to dehydrate faster than larger particles. Indeed, dehydration is typically studied using polycrystalline samples comprising nanoscale particles. Since we have demonstrated relatively rapid dehydration kinetics at ultralow temperatures, and using a comparatively enormous particle, our results represent an extreme case of the time taken for complete release of water. Therefore, $T(\text{on})$ could conceivably be even lower than -70°C , which strengthens our case even further. We have already addressed the difficulty of carrying out dynamic water vapour sorption measurements at very low temperatures. In our alternative approach we used a larger particle (i.e., single crystal) because the colour change in smaller crystallites at such low temperatures are difficult to quantify. Owing to the zero-order kinetics of dehydration, the *rate* of water release from **T1** is constant and we could thus use a larger particle (which would yield more reliable measurements) to demonstrate the principle. Since the kinetics of water release at temperatures just above $T(\text{on})$ would be extremely slow, we have relied on the Arrhenius parameters to determine the release time and to estimate $T(\text{on})$. We have added a statement to the discussion that more explicitly addresses the effect of particle size on the time required for total dehydration.

We believe that dehydration of zeolite nanoparticles at room temperature is quite different from dehydration of a comparatively large single crystal at -70°C . We are certainly not aware of any report in the literature of the dehydration of zeolites (even nanoparticulate zeolites) at temperatures below the normal freezing point of water, even if it is possible. Indeed, we agree with this referee (see last comment) that other materials will indeed be shown to exhibit similar dehydration behaviour to **T1**, and believe that our report will stimulate further studies in this regard.

Was the enthalpy of adsorption in the host determined?

Response: The most common method of determining enthalpy of adsorption ΔH_{ads} is to apply the Clausius-Clapeyron approach to isotherms recorded at several different temperatures. Our dynamic vapour sorption isotherms at 10, 25 and 40 are presented in Supplementary Figs. 12 to 14. Adsorption/desorption occurs in three distinct steps (regions E, F and G in Fig. 1) and, in principle, it is possible to determine the enthalpy of adsorption for each of these three zones. However, even across

the relatively wide temperature range of 30 °C (bearing in mind the limitations of commercial instrumentation), the ΔH_{ads} values reported here are only valid across the reported temperature-range in keeping with Kirchhoff's law¹ i.e. ΔH_{ads} should be reported as a function of temperature due to the difference in the partial molar heat capacities of the guest and the host (at constant pressure and composition) at different temperatures. Indeed, it is specifically the ability of **T1** to readily adsorb/desorb water across a wide temperature range that allowed us to use its colour as an indication of its hydration state. In response to the request by Referee 1 we now report ΔH_{ads} in the main text. Furthermore, a detailed discussion regarding the values of ΔH_{ads} with respect to loading has been included in Supplementary Text 5.3 of the ESI.

It is not clear why the authors did not perform their VT-SCXRD studies under equilibrium conditions at low partial pressures, to prevent water loss during experiments.

Response: Low partial pressures would promote water loss rather than preventing it; we have shown that the crystals dehydrate rapidly under low partial pressures/low relative humidity conditions as evidenced by TGA and optical microscopy. It was therefore necessary to seal the crystal to avoid water loss under the dry nitrogen stream of the cryostat. We also tried using a sealed capillary under humid conditions, but it was challenging to prevent icing.

The authors completely miss a discussion of the vast literature on water sorption and desorption in zeolites, mesoporous solids, MOFs and COFs.

Response: We agree completely with the referee that the literature on water sorption and desorption by a range of materials is vast. The list of different classes of water sorbing materials includes, but is not limited to, clays, zeolites, pharmaceutical and other molecular solids, MOFs, COFs, HOFs, salts, silica-based materials, carbon-based materials, glasses (e.g. perlite), organic polymers, etc. We feel that simply listing such materials would be extraneous without a discussion of their different behaviours, which would then take our introduction off on a tangent. Indeed, the strict word-count limitations for an article in *Nature* precludes such a discussion. In the process of preparing this manuscript it became clear to us that a comprehensive review of hygroscopic materials would be timely and extremely useful (it would have helped us immensely!). However, since some recent reviews of atmospheric water harvesting have discussed several classes of materials for water adsorption/desorption, we have added citations to two of them (Refs 7 and 8) for the interested reader.

Summarizing, the authors have shown that a channel host can absorb and desorb water, down to low temperatures. Such processes critically depend on the enthalpy of water ad/desorption, which can be controlled within a vast range via structure and functionalization of the host, and kinetically based on the physical size of the host material. There is really nothing surprising or original here.

Response: In this study, we have demonstrated for the first time the release of water by a material almost 100 °C below room temperature. This is surprising considering that water desorption below room temperature has rarely been reported (Supplementary Table 1), despite lowering the temperature of dehydration being one of the stated goals of the water harvesting and desiccant research communities. Although an immense amount of work has been undertaken to reduce water desorption temperatures using a variety of approaches (e.g., as the referee points out above, by downsizing crystallites to the nanoscale), 35 °C was very recently² described as an “ultralow temperature” for water release (See Supplementary Text 14). Not only have we shown that water

release is possible at sub-freezing temperatures, but we have established a new low-temperature benchmark for desorption by a hydrate (as given by T(on)).

Vast numbers of other materials exist whose interactions with water can be precisely tuned to reach similar behavior (the authors cite silica gel), including MOFs and COFs. It is surprising that the authors do not discuss in detail such other materials in their paper.

Response: Although it has not been reported, we also believe that similar behaviour (i.e. sub-freezing water release from crystal hydrates) will prove to be a more general phenomenon than previously thought. In fact, we had already included a discussion of other known hydrates that may potentially exhibit low temperature T(on) values (see Supplementary Text 14). However, we were not clear enough in the main text that this discussion could be found in the Supplementary Information. We have now clarified this in the main text. If you feel that this discussion needs to be moved to the main text, we would be happy to do so. However, it would add significantly to the word count.

Referee #2

on page 7 line 9, there should be an "is" before "also"

Response: This sentence has been reworded as part of our revision.

Though not a requirement for publication, it could be interesting to look at D2O / H2O exchange in these materials perhaps under Raman microscopy conditions. That would provide an alternative way to estimate water diffusion rates and mobilities in/out of the pores.

Response: We presume that the referee is suggesting that Raman microscopy could be used as an alternative imaging method similar to the optical microscopy experiments that we carried out to observe the progress of hydration and dehydration. This is an excellent suggestion but unfortunately we do not have access to a Raman microscope. However, we will try to establish a collaboration in order to undertake the suggested study in the future. In the same vein, we investigated the possibility of using computed tomography (details below), to which we do have access, as an alternative imaging approach. Such experiments could demonstrate a proof-of-concept approach to making similar observations for materials that do not change colour upon hydration and dehydration.

Computed Tomography: We postulated that it might be possible to observe a change in crystal density upon dehydration using computed tomography (General Electric Nanotom S). The calculated densities from SCXRD experiments carried out at 25 °C are 1.10 to 1.02 g cm⁻³ for **T1-R** and **T1-Y**, respectively. A single crystal was glued to a suitable support, mounted on the Nano-CT instrument and dehydrated under dry air flow (see figure below). A low-density plastic sleeve containing a small piece of dry cotton wool was placed over the crystal. A drop of water was added to the cotton wool and the crystal was left to equilibrate for 15 min under the now humid atmosphere. Although the crystal clearly changed colour from yellow to red, there was no observable difference between the CT images recorded for **T1-Y** and **T1-R**. Since the computed tomography did not yield any useful information, we have omitted it from the Supplementary Information.

NanoCT images of **T1-Y** (left) and **T1-R** (right) glued to a glass fibre. Inset: enlarged image with the crystal border outlined. The sleeve is clearly seen as a "shadow" and some condensation can be seen on the seam of the sleeve after 15 min of exposure to high humidity.

Referee #3

(commentary omitted)

P4, line 20: The fact that the crystal structure determined at 25 degC is isostructural to that at -173 degC does not rule out the possibility of a re-entrant phase transition.

Response: The 122 crystal structures of **T1** determined between -175 and -25 °C at 5 °C increments show no evidence of a phase transition of the framework in this range (see Supplementary Videos 8 to 11), either during cooling or heating. However, there is evidence of a thermal event during the cooling of **T1-R** in the DSC trace (Supplementary Fig. 55), which we interpret as partial ordering of water inside the channel, since it is not seen for **T1-Y**. The isostructurality of **T1** over a wide temperature range allowed us to use **T1-Y** as a structural reference for **T1-R** in terms of elucidating the influence of water on the host structure with changing temperature.

I am surprised that solid state ^1H NMR was not used to investigate this system, either for the enolimine to ketoenamine tautomeric equilibrium, or for the uptake/release of water, or for other characterization of the hydroxyl groups. Was this methodology considered and dismissed for some reason or was it overlooked?

Response: We attempted to study the enolimine to ketoenamine tautomeric equilibrium by means of solid-state NMR. However, these results yielded no further insight beyond what we had already obtained from the SC-XRD data. We therefore omitted the NMR experiments from the original version of our submission in the interests of brevity. We have now included them as Supplementary Text 7.3 of the ESI.

I am less surprised that spectroscopy in the IR spectral range was not employed (e.g., absorption or reflectance FT-IR; Raman), since interpretation might be more difficult. However I am wondering whether this was an omission by chance or whether the possibility of tracking O-H stretching frequencies and H-O-H bending/distortion modes was not considered valuable enough to attempt to collect and interpret data. Had the authors considered and dismissed the idea?

Response: We measured ATR-FTIR spectra at 0% and 90% RH and an additional spectrum of **T1** in a drop of liquid H₂O. Since these experiments provided no additional insight, we originally omitted them. However, details of the FTIR analyses are now included as Supplementary Text 7.4 of the ESI. Unfortunately, we do not have access to a Raman spectrometer.

References

1. Gatta, G. Della. Direct determination of adsorption heats. *Thermochim. Acta* **96**, 349–363 (1985).
2. Nguyen, H. L. *et al.* A Porous Covalent Organic Framework with Voided Square Grid Topology for Atmospheric Water Harvesting. *J. Am. Chem. Soc.* **142**, 2218–2221 (2020).

Reviewer Reports on the First Revision:

Referees' comments:

Referee #1 (Remarks to the Author):

Disappointingly, in their response to the comments of this reviewer the authors chose to basically not address the issues discussed. I am therefore repeating my previous review here (see below); all the points remain issues to be addressed. The statement that vapor sorption measurements at low temperature may be more difficult than at higher temperatures does not imply anything about the impact of such a study.

The serious concerns about the validity of $T(\text{on})$, lack of VT-SCXRD studies under equilibrium conditions, missing literature discussion and particularly lack of impact remain. Therefore, the conclusions expressed in the following text of the prior review remain as well. Publication in Nature cannot be recommended.

(prior review) The authors report the synthesis of a porous molecular organic crystal "T1" that can reversibly adsorb water into 1 nm channels (up to about 9 wt%), lined with some hydroxyl groups, at moderate relative humidity. As in many solvatochromic porous materials, the water sorption is accompanied by a color change (transition between 53 and 58% RH), which can be used to monitor uptake over a wide temperature range. The authors used a combination of different techniques (XRD, microscopy, DSC and molecular simulations) to ascertain that the confined water above -70 °C exists in a state of flux/disorder, which helps to allow dehydration of the hydrate to be observed due to weaker interactions than in bulk water.

Type V isotherms were observed, also pointing to weak host-water interactions. SCXRD data did not yield evidence for an ordered included water phase, but not surprisingly, enhanced electron density was found near the hydroxyl groups in the channels. Moreover, the kinetics of dehydration were determined down to temperatures below 0 °C – but does this not depend strongly on diffusion and thus on the size of the crystals? The Arrhenius-type activation energy for dehydration was found to be at the order of 40 kJ/mol, similar to that for the dehydration of silica gels.

Comments:

It is not clear why the authors find it particularly difficult to monitor desorption at lower temperatures – all that is needed to exclude atmospheric humidity is to perform the experiments under controlled gas atmosphere.

$T(\text{on})$ itself is not well defined – how long will you wait to see first traces of water emerge from the material? And which crystal size will you take as the reference size? For example, it has been shown that even hydrophilic zeolites can be dried at room temperature (instead of several hundred degrees) within short times when the crystals are present at the nanoscale. I would therefore argue that $T(\text{on})$ is by no means an intrinsic property of the materials – it strongly depends on size!

Was the enthalpy of adsorption in the host determined?

It is not clear why the authors did not perform their VT-SCXRD studies under equilibrium conditions at low partial pressures, to prevent water loss during experiments.

The authors completely miss a discussion of the vast literature on water sorption and desorption in zeolites, mesoporous solids, MOFs and COFs.

Summarizing, the authors have shown that a channel host can absorb and desorb water, down to low temperatures. Such processes critically depend on the enthalpy of water ad/desorption, which can be controlled within a vast range via structure and functionalization of the host, and kinetically based on the physical size of the host material. There is really nothing surprising or original here. Vast numbers of other materials exist whose interactions with water can be precisely tuned to reach similar behavior (the authors cite silica gel), including MOFs and COFs. It is surprising that the authors do not discuss in detail such other materials in their paper. As a result, the importance and potential impact of this work are far below the level expected for a paper in Nature. Publication is not recommended.

Reviewer #2:

No comments to the authors.

Author Rebuttals to First Revision:

RESPONSE TO REFEREES

Once again, we thank the referees for their careful evaluation of our manuscript.

Referee #1 (Remarks to the Author):

Disappointingly, in their response to the comments of this reviewer the authors chose to basically not address the issues discussed. I am therefore repeating my previous review here (see below); all the points remain issues to be addressed.

Response: We provided very detailed replies (a total of 1,500 words!) to the original comments by this referee, who has not specifically addressed the strengths or weaknesses of any of these responses. We are therefore unsure how to expand on our previous statements beyond what we have already stated.

The statement that vapor sorption measurements at low temperature may be more difficult than at higher temperatures does not imply anything about the impact of such a study.

Response: We believe that our statement about the difficulty of studying water sorption/desorption at low temperature is completely reasonable. Moreover, the experimental challenges are not the main focus of the work. In our view, the impact of the study relates to the observation of water loss by a crystal hydrate below the normal freezing point of water, and this point is well made. We are not aware of any existing report of this extremely surprising phenomenon.

The serious concerns about the validity of T(on), lack of VT-SCXRD studies under equilibrium conditions, missing literature discussion and particularly lack of impact remain. Therefore, the conclusions expressed in the following text of the prior review remain as well.

Response: We addressed these statements at length in our original response and it is unclear how to satisfy the referee without specific feedback.

We omit the rest of the referee's report since it repeats the original questions. Our responses to these have not changed.